# Neurodevelopment in Normocephalic Children Exposed to Zika Virus in Utero with No Observable Defects at Birth: A Systematic Review with Meta-Analysis

**DOI:** 10.3390/ijerph19127319

**Published:** 2022-06-14

**Authors:** Elena Marbán-Castro, Laia J. Vazquez Guillamet, Percy Efrain Pantoja, Aina Casellas, Lauren Maxwell, Sarah B. Mulkey, Clara Menéndez, Azucena Bardají

**Affiliations:** 1ISGlobal, Hospital Clínic—Universitat de Barcelona, 132 Rosselló Street, 08036 Barcelona, Spain; laia.vazquez@isglobal.org (L.J.V.G.); aina.casellas@isglobal.org (A.C.); clara.menendez@isglobal.org (C.M.); azucena.bardaji@isglobal.org (A.B.); 2Health Services Research Group, Vall d’Hebron Institut de Recerca (VHIR), Vall d’Hebron Hospital Universitari, Vall d’Hebron Barcelona Hospital Campus, Passeig Vall d’Hebron 119-129, 08035 Barcelona, Spain; efrain.pantoja@vallhebron.cat; 3Heidelberg Institute for Global Health, Heidelberg University, 69120 Heidelberg, Germany; lauren.maxwell@uni-heidelberg.de; 4Children’s National Hospital, Washington, DC 20010, USA; sbmulkey@childrensnational.org; 5Department of Neurology, The George Washington University School of Medicine and Health Sciences, Washington, DC 20037, USA; 6Department of Pediatrics, The George Washington University School of Medicine and Health Sciences, Washington, DC 20037, USA; 7Consorcio de Investigación Biomédica en Red de Epidemiología y Salud Pública (CIBERESP), 28029 Madrid, Spain; 8Centro de Investigaçâo em Saúde de Manhiça (CISM), Rua 12, Cambeve CP 1929, Maputo 1929, Mozambique

**Keywords:** Zika, neurodevelopment, language, cognitive, motor, delay, normocephalic

## Abstract

Zika virus (ZIKV) infection during pregnancy is a cause of pregnancy loss and multiple clinical and neurological anomalies in children. This systematic review aimed to assess the effect of ZIKV exposure in utero on the long-term neurodevelopment of normocephalic children born to women with ZIKV infection in pregnancy. This review was conducted according to the PRISMA guidelines for systematic reviews and meta-analyses. We performed a random effects meta-analysis to estimate the cross-study prevalence of neurodevelopmental delays in children using the Bayley Scales for Infant and Toddler Development (BSID-III). The risk of bias was assessed using Cochrane’s Grading of Recommendations Assessment, Development and Evaluation (GRADE) approach. Full-text reviews were performed for 566 articles, and data were extracted from 22 articles corresponding to 20 studies. Nine articles including data from 476 children found 6.5% (95% CI: 4.1–9.3) of infants and children to have any type of non-language cognitive delay; 29.7% (95% CI: 21.7–38.2) to have language delay; and 11.5% (95% CI: 4.8–20.1) to have any type of motor delay. The pooled estimates had a high level of heterogeneity; thus, results should be interpreted with caution. Larger prospective studies that include a non-exposed control group are needed to confirm whether ZIKV exposure in utero is associated with adverse child neurodevelopmental outcomes.

## 1. Introduction

Zika virus (ZIKV) infection during pregnancy is a cause of pregnancy loss and multiple clinical and neurological birth defects in children, including microcephaly and congenital Zika syndrome (CZS) [1,2,3]. Vertical transmission of the virus occurs in 20% to 30% of infants born to ZIKV-infected pregnant women, and the range of manifestations is very wide, from pregnancy loss to CZS [4,5]. Almost half of infants infected through vertical transmission do not present with any ZIKV-associated signs or symptoms within the first week of life [4,5]. Fetuses of infected women have a 5 to 14% risk of developing CZS and a 4 to 6% risk of microcephaly, and in those who develop CZS, 80% of them present with microcephaly [5]. Microcephaly is usually readily apparent at birth, but some adverse consequences of in utero ZIKV exposure may only be observable in early childhood or when children start school, which is still critical to know [6]. Recent studies have shown that antenatal ZIKV exposure is associated with cognitive and language delays in normocephalic children [1,6,7,8,9]. The first article reporting an alteration in neurodevelopment revealed that 40% of normocephalic children exposed to ZIKV in utero had some type of neurodevelopmental anomaly, with the most frequently reported delay being in language development [10]. Subsequent publications have also found that language function was the most affected domain in children whose mothers tested positive for ZIKV during pregnancy and who were normocephalic at birth [6,9,10].

ZIKV infection is assessed through molecular and serologic assays and may be subject to confirmatory testing to distinguish between ZIKV and endemic arboviruses with clinically similar presentations such as dengue virus (DENV) and chikungunya virus (CHIKV). Most ZIKV infections are asymptomatic, and women who are tested for ZIKV may present outside of laboratory detection sensitivity window, which can make it difficult to confirm infection. After birth, neonatal laboratory screening poses additional challenges, since infants with antenatal exposure may have negative serology tests and infants that were not exposed may test positive due to maternal IgG antibodies that crossed the placenta [6]. This represents an additional diagnostic challenge due to the unavailability of resources, especially in low- and middle-income countries (LMIC) where the ZIKV outbreaks occurred, leaving most of the ZIKV-exposed children without laboratory evaluation at birth [6]. In addition, ZIKV-associated birth defects and neurodevelopmental abnormalities have been reported in children who have tested negative for ZIKV at birth [7]. Children who are exposed to ZIKV in utero can develop postnatal microcephaly or other signs of CZS that were not observable at birth [11]. Children who are normocephalic at birth or who did not present with any anomalies on fetal or postnatal cranial ultrasound or MRI may present with neurodevelopmental delay sometime later in childhood [8]. The early identification of abnormal neurodevelopmental outcomes in children is central to providing optimal treatment to try and improve cognitive, social and behavioral functioning [7]. There are more than 100 childhood developmental assessment tools (CDATs) that are used to evaluate neurocognitive development in infancy and early childhood, including The Guide for Monitoring Child Development, Parents’ Evaluation of Developmental Status (PEDS), Ages and Stages Questionnaire (ASQ), and the Bayley Scales for Infant and Toddler Development (BSID-III) [12,13]. CDATs are meant to be applied at different ages and cover a number of important developmental domains, including social and emotional, language, cognitive (learning, thinking, problem-solving), and movement-physical development. CDATs have been developed and tested in different populations and languages, but they have different validity, reliability, cultural adaptability, accessibility, need of training, administration time, geographical uptake and clinical relevance and utility [13]. Some tools, such as the Denver Prescreening Developmental Questionnaire, Access Portfolio, and Abbreviated Developmental Scale-I (EAD-1), require highly specialized training, which may limit their usage [13].

This systematic review was conducted based on the lack of evidence on health effects of children born from ZIKV infected mothers, with no observable birth defects. This systematic review aims to assess the effect of ZIKV exposure in utero on the long-term neurodevelopment of normocephalic children born to women with ZIKV infection in pregnancy, whether or not infants tested positive for ZIKV infection at birth.

## 2. Materials and Methods

### 2.1. Search Strategy and Selection Criteria

This systematic review was conducted according to the PRISMA guidelines for Systematic Reviews and Meta-Analyses [14]. The systematic search of articles was conducted using a combination of text terms, which were tailored for application in the following databases: MEDLINE, Scopus, The Cochrane Database of Systematic Reviews, and the Cochrane Central Register of Controlled Trials. Databases were searched from inception to 1 March 2021 in any language from any geographic location, and Rayyan software was used to perform database merges, the deletion of duplicate citations, and title abstract and full-text screening [15]. The search string included the following text terms: ((Zika OR ZIKV) AND (Child OR children OR infant) AND (neurodevelopment OR neurodevelopmental OR development OR neurodevelopmental OR Bayley OR BSID OR delay OR language OR motor OR cognitive)); see Appendix A. Systematic reviews and meta-analyses were manually excluded. Grey literature sources were not assessed. The question under investigation was, ‘What is the prevalence of delays in motor, cognitive, and language function among children whose mothers tested positive for ZIKV during pregnancy and who were normocephalic at birth?’. In this article, the definition of “cognitive domain/delay” is based on the neurodevelopmental tool used. While “language” is an aspect of cognition, when we mention “cognitive”, we refer to “non-language domain/delay”.

The systematic review searched for published articles in which the study population included normocephalic infants and infants born to mothers with ZIKV infection during pregnancy, whether or not the infants were ZIKV tested at birth. Studies including only data from children with microcephaly, or other ZIKV-associated adverse outcomes or conditions that were observable at birth, were excluded. Studies that also included data on children with no observable defects are birth were included. 

For the meta-analysis, additional exclusion criteria were applied, including (1) studies reporting neurodevelopment with tools other than BSID-III, and (2) studies not reporting data for the three different domains evaluated and by type of results (severe delay, moderate delay, mild delay, normal results, above normal results). 

Two review authors independently assessed the eligibility of the studies identified in the search and disagreements were resolved by a third reviewer. The citations of included studies were reviewed to identify additional studies for inclusion. The study protocol was registered in PROSPERO (CRD42021242262) prior to the initiation of the search [16].

### 2.2. Data Extraction

Information was extracted by two reviewers independently, and differences in data extraction were resolved through consensus. Data items collected included articles’ identification (first author, title, link and date of publication, and journal), study design, type of population included, the ascertainment of maternal and infant ZIKV status, control group and other study design-related issues, type of developmental tool including the domains assessed, assessor training, ages of assessment, main objective of the study, main findings, and, in the case of articles using the BSID-III, point estimates and associated uncertainty measures for motor, cognitive, and language scores for −1 to −2 SD and below −2 SD (moderate and severe delay). Unclear or missing information in the articles was entered as “Not reported”.

### 2.3. Risk of Bias, Quality Assessment, Data Extraction and Analysis

For the studies included in the meta-analysis, two reviewers independently assessed the risk of bias and the quality of the non-randomized studies (certainty of the evidence), using The Risk of bias in non-randomized (ROBINS-I) tool from Cochrane Scientific Committee [17]. Data were recorded in the online GRADEpro Guidelines Development Tool (GDT) [18], following the GRADEpro Guideline Development Tool Guidelines [19]. The assessment of the risk of bias included the analysis of (1) confounding, (2) selection bias, (3) deviations from intended intervention, (4) measurement of outcome, (5) outcome definition, and (6) others (inconsistency, indirectness, imprecision, publication bias, large effect, and possible confounding). Each of the aforementioned criteria were assessed as presenting a very low, low, unclear, or high certainty of the evidence. Articles presenting two or more assessments of high risk (illustrated in red) were considered of very low certainty of evidence. Articles presenting one assessment of high risk (in red) were considered of low certainty of evidence. Articles presenting one assessment of uncertain risk (in orange) were considered of moderate certainty of evidence. Articles presenting all the assessments of low risk (in green) were considered of high certainty of evidence.

### 2.4. Synthesis Methods

The principal endpoint of the meta-analysis was the domain-specific prevalence of cognitive, motor, and language neurodevelopmental delays, defined as mild (−1 to −2 SD), moderate (−2 to −3 SD), or severe (>−3 SD) based on the number of standard deviations (SD) below the mean global scores estimated with BSID-III. The categorization of any delay (including mild, moderate, and severe delay) (<−1 SD), and moderate and severe delay (<−2 SD) was performed with studies that included disaggregated data for every category in the BSID-III scale. Pooled prevalence estimates with corresponding 95% confidence intervals (CI) were calculated using Freeman–Tukey double arcsine transformation [20]. Cross-study heterogeneity was estimated using Cochran’s Q statistic [21] and the I^2^ index. For I^2^ between 50 to 75% (included), a random-effects model based on the DerSimonian and Laird method was used. If heterogeneity was greater than 75%, we reported study results narratively, without a pooled estimate. The pooled analyses were performed using Stata 16 (StataCorp. 2019. Stata Statistical Software: Release 16. College Station, TX, USA: StataCorp LLC.) [22].

## 3. Results

### 3.1. Study Selection and Characteristics

The PRISMA Flow chart (Figure 1) depicts the study selection and screening process. 

Our search identified 812 articles in English, Spanish, Portuguese or French. One additional article was identified through reviewing the references of included studies. After the removal of 246 duplicates, 22 of the 566 articles included in the title–abstract screening were included in the review [2,7,8,10,23,24,25,26,27,28,29,30,31,32,33,34,35,36,37,38,39]. No full texts were excluded. Data were extracted from the 22 eligible articles, which corresponded to 20 studies. Two articles included in this review were performed with children from the same study from different time points. Characteristics of the 22 articles included in the systematic review are described in Table 1. 

Studies were heterogeneous in design, the inclusion, or not, of a control group, the definition and ascertainment of maternal ZIKV infection, the definition and ascertainment of congenital ZIKV exposure, inclusion and exclusion criteria for infants and children, length of follow up, location, and type of tools used for neurodevelopmental assessment. Although there were differences in the description of the study populations across articles, all children included in the systematic review were normocephalic, without any apparent ZIKV clinical anomalies either screened or detected at birth (abnormal brain imaging, including structural and nonstructural abnormalities, calcifications and cysts, was detected in children from two studies). The included articles used data from retrospective (*n* = 4) [2,7,23,24], and prospective (*n* = 10) cohort studies [8,10,25,26,27,28,29,30,31,32], cross-sectional (*n* = 4) [33,34,35,36], and case series (*n* = 4) [36,37,38,39]. The median age of neurodevelopment assessment was 14 months, and four studies included assessments more than once during children development [23,24,31,37]. These four studies included between two and four assessments at birth or months 2, 4, 6, 12, 15, 18, and 25.

### 3.2. Maternal ZIKV Ascertainment

Maternal ZIKV status ascertainment included: RT-PCR [10,24,25,26,27,28,29,32,33,34,36,37,38,39], RT-PCR or serological tests [2,7,23,30,31,35,40], and serology [8]. RT-PCR (14 articles) [10,24,25,26,27,28,29,32,33,34,36,37,38,39]; laboratory confirmed and probable (e.g., symptomatic during outbreak) cases (7 articles) [2,7,23,30,31,35,40], and probable cases, by positive anti-ZIKV IgG at delivery (1 article) [8]. Some articles reported data on symptomatic pregnant women, who usually reported a rash (8 articles) [2,10,24,25,26,30,31,36], while others included pregnant women with a laboratory confirmed infection, irrespective of symptoms. Seven articles included control groups of ZIKV unexposed children [25,28,29,31,33,34,35]. 

### 3.3. Infant ZIKV Ascertainment

Five articles tested newborns at birth for ZIKV infection by RT-PCR in different sample specimens (neonatal blood, urine, cord blood or placental blood) [2,23,37,39,40]. One study reported screening for ZIKV in infants born with CNS anomalies [32].

### 3.4. Covariate Ascertainment

Eight articles reported data on testing for other congenital infections in maternal or neonatal samples [23,27,31,33,36,37,39]. Eleven articles reported data on the socio-economic characteristics of the families included in the studies [8,24,28,29,31,33,35,36,37,40]. There were differences in the way studies reported socio-demographic data. Most studies reported proxy variables related to the social determinants of health, such as maternal educational status [8,28,29,31,33,35,36,40]. Other studies reported on individual or household economic indicators, including: “household income” [8], living with monthly minimum wage [31], or average income [36].

### 3.5. Neurodevelopmental Screening Tool Outcomes

There were 11 different neurodevelopmental assessment and screening tools used to evaluate developmental outcomes in normocephalic children who were exposed to ZIKV during pregnancy (Appendix A). Five tools measured one neurodevelopmental domain: Prechtl’s General Movement Assessment (motor), The Alberta Infant Motor Scale (gross motor), The Fagan Test of Infant Intelligence (cognitive), The Modified Checklist for Autism on Toddlers (behavior), and The French MacArthur Inventory Scales (French language). All other screening and assessment tools included multiple domains such as cognitive, communication, problem solving, personal–social, cognition, motor, etc. All studies evaluated child neurodevelopment in the first two years of age, with the exception of one study in which the BSID-III scale was applied at 38 months of age [38], and two studies in which a second evaluation of the BSID-III scale occurred after the second year of age [2,27]. Most articles in which neurodevelopment was assessed by tools other than the BSID-III concluded that delays were present in the population under study (Table 2). 

Those delays included absence of fidgety movements [39], neurodevelopmental abnormality [7], abnormalities by physical examination [31], delayed receptive language [35], gross motor and language delays [23], abnormal developmental outcomes (communication, social cognition, and mobility) [30], abnormal neurodevelopmental performance in language and motor domains and poor visual recognition memory [29], delayed milestones [36], and neurodevelopmental anomalies [24]. The BSID-III was assessed in the first (three articles), second (three articles), and third year of life (one article), and within a range of months after the first year (four articles). In studies that assessed development with the BSID-III in children with prenatal ZIKV exposure (476 children evaluated from nine articles), the pooled prevalence of any cognitive delay was 6.5% (95% CI: 4.1, 9.3; *n* = 41 children; I^2^ = 31.0%, differences between publications were low). The prevalence of any language delay was 29.7% (95% CI: 21.7, 38.2; *n* = 148 children; I^2^ = 64.9%, differences between publications were large). The prevalence of any motor delay was 11.5% (95% CI: 4.8, 20.1; 77 children; I^2^ = 78.4%, differences between publications were large) (Figure 2 and Table 3). All the data from specific studies and the pooled prevalence can be found in Appendix A.

The pooled prevalence of moderate-to-severe delay (<−2 SD) was estimated in eight of the nine studies that used the BSID-III scales, because one article did not provide disaggregated data for mild (<−1 to <−2 SD), moderate (<−2 SD to <−3 SD), or severe delay (<−3 SD). In those eight articles, the pooled prevalence of moderate and severe cognitive delay was 1.9% (95% CI: 0.4, 4.1), for moderate and severe language delay was 8.4% (95% CI: 5.4, 11.8), and for moderate and severe motor delay was 2.2% (95% CI: 0.6, 4.5) (Figure 2 and Table 4).

### 3.6. Risk of Bias and Quality Assessment

Details on the information obtained in the assessment of the risk of bias and quality of included studies can be found in Appendix A. There were seven articles presenting at least one category of high risk; thus, certainty of the evidence was very low. There were two articles with at least one category of uncertain risk, and thus of moderate evidence. The main reason for the low quality in the outcome assessment was the very small sample size. There were also differences in the exposure and outcome ascertainment and high levels of heterogeneity in the populations under study. Absolute sample size is small, and these were very select populations (prenatal ZIKV exposure, normocephalic and no apparent ZIKV birth defects, and long-term follow up). The studies contributing a higher sample size to the pooled prevalence were those from Nielsen-Saines K. et al. (*n* = 146) [26], Lopes Moreira M.E. et al. (*n* = 94) [10], and Peçanha P.M. et al. (*n* = 84) [37]. The studies with the highest level of uncertainty were Sobhani N.C. et al. (*n* = 3) [27] and Abtibol-Bernardino M.R. et al. (*n* = 26) [38]. In addition to unmeasured confounding, difficulties and differences in maternal ZIKV ascertainment, including uncertainty in gestational age at infection, lack of and differences in ascertainment of socioeconomic status and other markers of vulnerability, and differences in the timing and tools used for outcome ascertainment contributed to the high level of cross-study heterogeneity.

## 4. Discussion

This systematic review assessed the effect of ZIKV exposure in utero on the long-term neurodevelopment of healthy-at-birth normocephalic children born to women with ZIKV infection during pregnancy based on laboratory criteria. To our knowledge, this is the first systematic review and meta-analysis to assess the mid- and long-term effects of prenatal exposure to ZIKV on normocephalic children’s neurodevelopment in the first two years of age. This systematic review included published data from observational prospective and retrospective cohorts, cross-sectional studies, and case series, reporting on children prenatally exposed to ZIKV who were normocephalic and asymptomatic at birth, whether or not they were ZIKV-infected at birth. Many different tools were used to assess the neurodevelopment of ZIKV prenatally exposed children, with the BSID-III scales being the most common assessment used in the articles included in this review (*n* = 11/22). Articles assessing neurodevelopmental outcomes with tools other than BSID-III had different study designs, but the same objective and similar findings [41]. There is concurrent validity of the WIDEA with the Bayley, and this has been recently reported [41]. Prechtl’s method for GMA, particularly in the fidgety period, can be useful for the early assessment of ZIKV-exposed infants and provide substantial contributions to identify those who might benefit most from early intervention [39]. While most of the articles found developmental delays in ZIKV exposed children, two studies with a total of 250 prenatally exposed children did not find any association between presumed ZIKV exposure during pregnancy compared to 125 matched controls [28,34]. One study of 94 children born to women with positive ZIKV RT-PCR and 46 children from a neurotypical control group, conducted in Brazil, reported that ZIKV-exposed children and neurotypical non-exposed controls had similar frequencies of developmental delays (cognitive, language, motor and social-emotional), assessed by The Survey of Wellbeing of Young Children (SWYC) at 28 months of age [34]. However, the authors acknowledged that if a more comprehensive developmental assessment tool, such as the BSID-III, had been used, a higher percentage of children with delays may have been identified [34]. Moreover, the authors suggested that their results could not exclude later-onset neurodevelopmental repercussions, and that a more comprehensive tool such as the BSID-III should be performed in these children [34]. Similarly, in a large population-based mother–child cohort of 156 ZIKV-exposed and 79 unexposed children who were normocephalic at birth conducted in Guadalupe, Martinique, and French Guiana, no statistically significant associations were found between maternal ZIKV exposure status and motor, communication, personal–social and problem-solving outcomes evaluated at 24 months of age [28]. ZIKV-exposed and -unexposed groups presented similar levels of motor development, problem solving, and personal–social skills, and the language domain was most negatively affected in ZIKV-unexposed compared to exposed children (20.3% vs. 8.3%) [28]. However, as the authors suggested, a high loss to follow-up occurred in the control group, which could have affected results, since families with children with developmental concerns could be more prone to continue in the study [28]. The main limitation of this study was the definition of the control group, which included a negative serology for anti-ZIKV IgG in mothers at delivery, but women could have been infected during pregnancy and not have enough IgG antibodies at delivery, or false negative results could have played a role [28]. There were only seven studies that included a control group of ZIKV-unexposed children. For studies that assessed neurodevelopment using the BSID-III, our pooled prevalence analysis added evidence to support the finding of neurodevelopmental delay in normocephalic children with in utero ZIKV exposure, especially in the domain of language function, with 30% of cases having a low score, which, in 8.4% of cases, was moderate to severe. From the 11 studies evaluated, only one study did not find a neurodevelopmental delay in ZIKV exposed children (Rodrigues Gerzson L. et al. [26]). This study included a control group of sex- and age-matched normocephalic children with no maternal history of ZIKV or other congenital infection. Both groups had similar socio-economic backgrounds, which might have played a role in the lack of association. In this study, it was not clear if the control group had laboratory confirmation of negative ZIKV infection or which type of laboratory test was used [26]. Additionally, false negative cases could have been included in the control group, given that all women were living in endemic areas for arboviruses, that many ZIKV infections are asymptomatic, and that there were limitations of available diagnostic tools in the study geographic areas.

Recent studies have found language delays in children who were exposed to ZIKV during pregnancy, but a causal association has not yet been established [42]. The timing of ZIKV infection, route of infection, maternal and fetal immune status, socioeconomic factors (including gender roles, parental educational level, origin, migration history, ethnicity), the public health response to the epidemic at local, national and the international level, amongst other biological and social factors and exposures may influence the relation between maternal ZIKV infection and adverse fetal, infant, and child outcomes. The incidence of both ZIKV and microcephaly are highest in populations in under-resourced settings, which may have higher levels of exposure to infectious disease, environmental and workplace pathogens, dietary deficiencies, and fewer resources to detect or prevent congenital infections and to interrupt affected pregnancies [43,44]. In most studies, socio-economic data to inform demographic characteristics of the population under study were collected, including those studies with a control group, and showed that affected families and matched controls did not differ on such baseline characteristics, thus minimizing the potential confounding between socioeconomic status and developmental outcomes. Most studies argued that socioeconomic characteristics were not different between children enrolled in the study and the source population, but data were not provided to substantiate this assumption. The lack of a socioeconomic assessment in the included studies did not allow us to properly assess the socioeconomic status of the children as a potential confounding factor. However, the main concern was not merely whether the children were from similar socioeconomic statuses, but the fact that the socioeconomic factors might play a role in creating inequalities related to risk of ZIKV infection, lack of access to preventive and clinical care, lack of early educational exposures, and thus a higher risk of poor developmental outcomes, especially for children from the poorest-resource settings and families. Additionally, limitations in laboratory screening, access to antenatal care, study recruitment, and fetal diagnosis are also relevant factors that might play an important role. 

Our results have important implications for practice and policy. The close monitoring of ZIKV prenatally exposed children is needed throughout childhood to allow for the early detection of developmental impairment and to inform subsequent specific clinical care (language, motor, or cognitive therapies). Further research is needed to understand whether there is a significant association between prenatal ZIKV exposure and language delay, or if this association is confounded by socioeconomic characteristics of study populations. The main limitations of the review were inherent limitations of the studies that were included, and the assessment tools. The ascertainment of gestational age at infection, and ZIKV exposure, is challenging given the high proportion of asymptomatic ZIKV infections. All included studies were affected by selection bias. This is because the accuracy of RT-PCR is related to the time to infection, and symptomatic women were more likely to present closer to the infection. In addition, most studies did not include a control group. The lack of a control group in many studies limited the ability to compare the prevalence of the delay with a similar population of children not exposed to ZIKV in utero. There is limited funding and an evolving understanding of the etiology of disease, which have complicated efforts to have a control group of unexposed pregnancies in an endemic area with the transmission of several arboviruses and where infection may be subclinical. Heterogeneity among studies for ZIKV diagnosis in mothers (by RT-PCR, serological methods, reporting of symptoms, etc.), the inclusion of women with laboratory-confirmed and probable/suspected cases, and the inclusion of infected children and children who were not tested might bias our results. For that reason, this review only included pregnancies with ZIKV infection by laboratory diagnosis, which is an underrepresentation of all the cases of women with ZIKV infection during pregnancy. The outcome of interest in this review (apparently healthy at birth) was also challenging to be confirmed since there was variability in the postnatal evaluation of infants across the studies. First, because this group has especially high lost-to-follow-up rates, and second, because of a lack of standardized diagnostic tools at birth used throughout the studies, such as neuroimaging with head ultrasound or brain magnetic resonance imaging or ZIKV laboratory testing, this might have increased the number of children in this group, who could actually have CZS or other ZIKV-associated birth defects.

The heterogeneity of neurodevelopmental assessment tools used hindered the inclusion of half of the studies in the meta-analysis. The small number of studies, and children included, further limited the possibilities to perform different analyses comparing those children born to confirmed women vs. those born to probable/suspected mothers, or infected children vs. negative children or those not tested. Another issue to consider is the inherent limitations of the tools for child neurodevelopmental evaluation. For example, the recommendation of the BSID-III to define developmental delay in children with scores −1 SD is a subject of debate. Wider cut-off points may be important for referral for neurostimulation [17]. In this sense, an underrepresentation of the real burden of children with developmental delay due to congenital exposure to ZIKV could result from studies using the BSID-III, as well as an under-identification of mild cases in studies utilizing other tools. Lastly, this study presents data from published articles; however, some children from the same study population included in different articles might have been counted twice in the meta-analysis. Based on these limitations, the results should be interpreted with caution.

## 5. Conclusions

This systematic review concludes that ZIKV during pregnancy is a risk factor for early childhood neurodevelopmental delay in normocephalic children. This is the first study assessing the pooled prevalence of neurodevelopmental delays estimated in studies of normocephalic children prenatally exposed to ZIKV. Seven of the nine studies assessed in the meta-analysis had a very low certainty of the evidence, and two of them had a moderate certainty. Articles reviewed conclude that the language domain was the most negatively affected area, impacting one-third of normocephalic children with prenatal ZIKV exposure. Using the BSID-III scale, almost 30% of children presented any language delay, and 8.4% moderate-to-severe language delay, while moderate and severe delays were lower in the motor and cognitive functions. As these children grow older, further studies will be essential to unveil whether these developmental delays continue into adolescence and what factors may accelerate or slow the rate of delay. Delayed child neurodevelopment might be due to different factors, such as nutrition or socioeconomic factors. Those factors might confound the association of ZIKV prenatal exposure with neurodevelopmental delays, and the adjustment for these factors, as through matched controls, should be included in future studies. Due to the lack of a control group in many included studies, this study could not confirm nor discard the association between prenatal ZIKV exposure and delayed child neurodevelopment. High loss to follow-up rates, difficulties in the ascertainment of the timing or presence of ZIKV infection in pregnancy, selection bias in studies limited to symptomatic pregnant women, and difficulties in inferring fetal exposure from maternal exposure are all major issues to be considered in analyzing ZIKV maternal–infant studies and the development of further projects. Larger prospective studies including non-exposed control groups are needed to confirm whether antenatal ZIKV exposure is associated with delayed child neurodevelopment.

## Figures and Tables

**Figure 1 ijerph-19-07319-f001:**
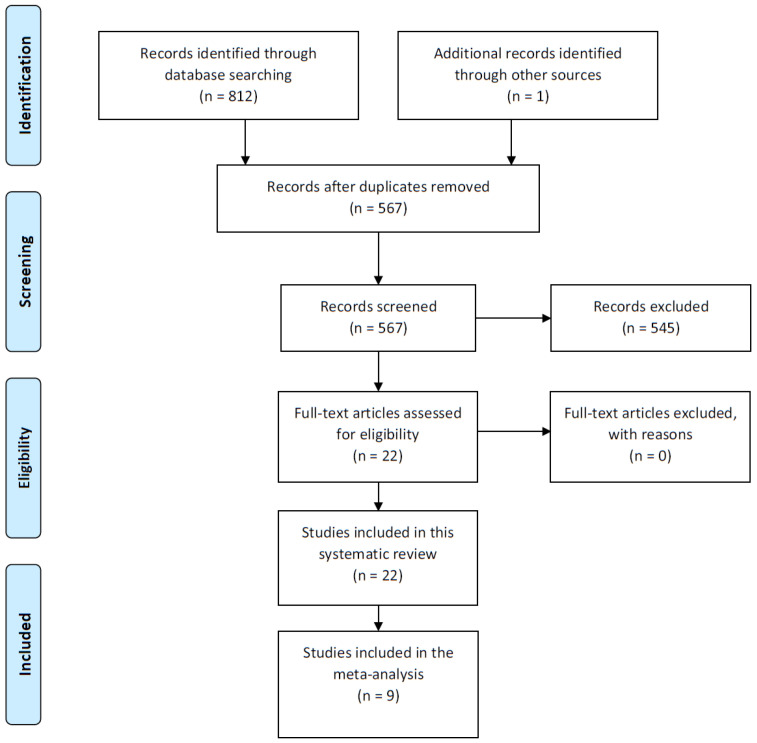
Systematic review PRISMA flow diagram.

**Figure 2 ijerph-19-07319-f002:**
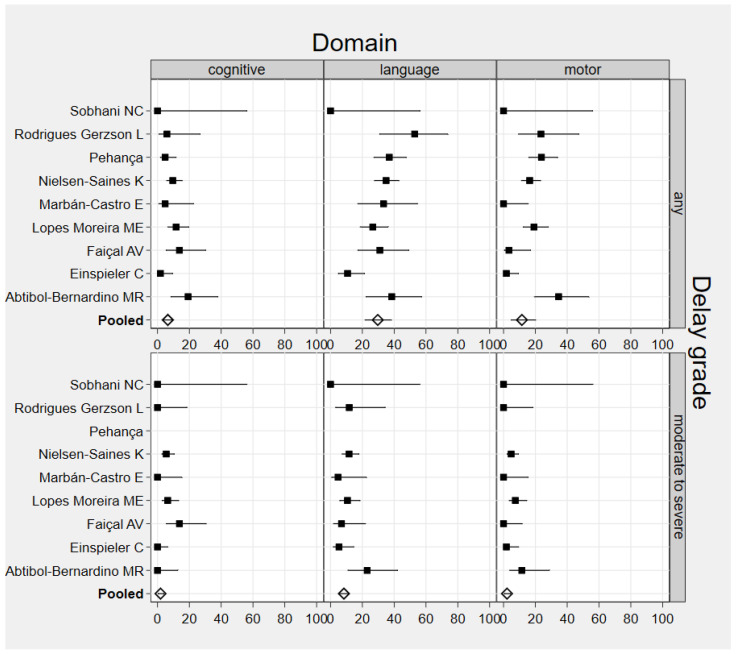
Forest plot of the prevalence of neurodevelopmental delays in children with prenatal ZIKV exposure and study authors. I²: variation in prevalence attributable to heterogeneit.; Cognitive domain: Differences between publications are low (I^2^ = 47.8%); thus, fixed-effects pooled prevalence was considered. Language domain: Differences between publications are low (I^2^ = 0.0%); thus, fixed-effects pooled prevalence was considered. Motor domain: Differences between publications are low (I^2^ = 17.5%); thus, fixed-effects pooled prevalence was considered.

**Table 1 ijerph-19-07319-t001:** Characteristics of studies included in the systematic review.

First Author	Countries (*n* of Children Included)	Study Design	Neuro-Developmental Tool	ZIKV Status-Related Inclusion Criteria (Pregnant Women) *	Maternal and Infant-Status Related Inclusion Criteria (Infants)	Children Age at Neurodevelopmental Assessment (Number of Assessments)	Control Group
**Lopes Moreira ME**	Brazil (94)	Prospective observational cohort study	BSID-III	Confirmed by RT-PCR	Born to ZIKV-confirmed mothers. Included newborns with abnormal brain imaging (structural and nonstructural abnormalities), but none were ZIKV tested	Between 12 and 18 months (1)	No
**Einspieler C**	Brazil (56)	Prospective observational cohort study	BSID-III	Confirmed by RT-PCR	Born to ZIKV-confirmed mothers. Infants not tested for ZIKV	12 months (1)	Yes, sex- and age-matched neurotypical controls without exposure to maternal ZIKV
**Nielsen-Saines K**	Brazil (146)	Prospective observational cohort study	BSID-III	Confirmed by RT-PCR	Born to ZIKV-confirmed mothers. Infants not tested for ZIKV	12 months (1)	No
**Rodrigues Gerzson L**	Brazil (17)	Cross-sectional study with control group	BSID-III	Confirmed by RT-PCR	Born to ZIKV-confirmed mothers. Infants not tested for ZIKV	18 months (1)	Yes, sex- and age-matched normocephalic children with no maternal history of ZIKV or other congenital infection.
**Faiçal AV**	Brazil (29)	Prospective child cohort	BSID-III	Probable by positive IgG at delivery	Infected children confirmed by PCR, but normocephalic	19 months (1)	No
**Peçanha PM**	Brazil (84)	Longitudinal exploratory case series	BSID-III	Confirmed by RT-PCR	Born to ZIKV-confirmed mothers. Infants not tested for ZIKV	9 and 15 months (2)	No
**Sobhani NC**	Brazil (3)	Prospective observational cohort	BSID-III	Confirmed by RT-PCR	Born to ZIKV-confirmed mothers. Infants not tested for ZIKV	25 months (2 children), and at 39 months (one child) (1)	No
**Cranston JS**	Brazil (112)	Retrospective cohort of women and prospective cohort of children	BSID-III	Confirmed/probable (symptom referral, or abnormal US findings, or positive laboratory assay)	Born to ZIKV-confirmed mothers. Infants not tested for ZIKV	6 to 42 months (1)	No
**Coutinho CM**	Brazil (199)	Prospective population-based cohort study	BSID-III	Confirmed by RT-PCR	Born to ZIKV-confirmed mothers. Infants not tested for ZIKV	3 months (1)	No
**Abtibol-Bernardino MR**	Brazil (26)	Case series	BSID-III	Confirmed by RT-PCR	Born to ZIKV-confirmed mothers. Infants not tested for ZIKV	38 months (1)	No
**Marbán-Castro E**	Spain (21)	Prospective cohort **	BSID-III	Confirmed (by PCR) and probable (by serological methods and microneutralization)	Born to ZIKV-confirmed or probable mothers. Three children presented abnormal brain findings (calcifications or cysts). Neonates were negative for ZIKV screening in placenta, cord blood and neonatal blood and urine	24 months (1)	No
**Soares-Marangoni DA**	Brazil (2)	Case report	Prechtl’s GM assessment and AIMS	Confirmed by RT-PCR	Born to ZIKV-con-firmed mothers. Infants not tested for ZIKV	4 and 12 months (1)	No
**Rice ME**	U.S. territories and freely associated states (1386)	Retrospective analysis of medical/surveillance collected data	Validated screening tools recommended by the American Academy of Pediatrics ***	Confirmed and probable (laboratory evidence of confirmed or possible ZIKV infection)	Born to ZIKV-confirmed and probable mothers. Infants not tested for ZIKV	12 months (1)	No
**Oliveira Vianna RA**	Brazil (82)	Longitudinal observational study	DDST	Confirmed and probable/suspected (women with rash during pregnancy or three months before pregnancy coinciding with the PHENC in Brazil)	Group 1: Born to ZIKV-confirmed mothers; Group 2: Born to ZIKV-negative mothers; and Group 3: Born to mothers not tested for ZIKV but who tested negative for other congenital infections. Infants not tested for ZIKV	6, 12 and 18 months (3)	Yes, 26 children whose mothers tested negative by RT-qPCR for ZIKV (Group 2)
**Valdes V**	Puerto Rico (65)	Cross-sectional study	MSEL (translated to Spanish and adapted for Puerto Rico)	Confirmed/probable/suspected (by PCR or serology)	Born to ZIKV-confirmed mothers. Infants not tested for ZIKV	3 to 6 months; or 9 to 12 months (±2 weeks) (1)	Yes, 36 children born from mothers with negative PCR or ELISA for ZIKV
**Lee EH**	USA (148)	Retrospective analysis of medical/surveillance collected data	No test specified. Neurodevelopmental abnormalities possibly associated with ZIKV.	Confirmed and probable (Laboratory evidence of ZIKV infection during pregnancy)	Born to ZIKV-confirmed or probable mothers. Infants not tested for ZIKV	At birth, 2, 6, and 12 months (4)	No
**Mulkey SB**	Colombia and USA (70)	Prospective cohort	WIDEA and AIMS	Confirmed and probable (CDC clinical criteria for probable ZIKV infection and laboratory evidence of ZIKV confirmed by one or more tests, including PCR, IgM, IgG, and PRNT)	Born to ZIKV-confirmed mothers. Infants not tested for ZIKV	One or two assessments between 4 and 18 months (1–2)	No
**Da Silva PFS**	Brazil (140)	Cross-sectional study, nested in a cohort study	SWYC	Confirmed by RT-PCR	Group 1: Severe microcephaly; Group 2: Moderate microcephaly; Group 3: Prenatal ZIKV exposure confirmed by maternal RT-PCR testing but no microcephaly; Group 4: Neurotypical control group. For this review, only Groups 3 and 4 were considered. Infants not tested for ZIKV	28 months (1)	Yes, 46 neurotypical children with neither microcephaly nor any other brain abnormalities detectable by brain US at birth who were born to mothers with no laboratory evidence of ZIKV infection during pregnancy (Group 4)
**Familiar I**	Mexico (59)	Prospective cohort	MSEL and FTII	Confirmed by RT-PCR	Born to mothers with confirmed infection, normocephalic and asymptomatic. Infants not tested for ZIKV	6 months (1)	Yes, 45 healthy children without ZIKV exposure
**Cabral Maia AMP**	Brazil (17)	Cross-sectional case series study	None (Child Health Booklet developed by the Brazilian Ministry of Health)	Confirmed by laboratory (diagnostic tool not reported)	Born to ZIKV-confirmed mothers. Infants not tested for ZIKV	10–25 months (1)	No
**Pimentel R**	The Dominican Republic (42)	Retrospective cohort analysis of children	DDST	Confirmed by RT-PCR	Born to ZIKV-confirmed mothers. Infants not tested for ZIKV. Only neonates with microcephaly were included	1, 2, 3, 6, 9, 12, 15, and 18 months (8)	No
**Grant R**	Guadeloupe, Martinique, and French Guiana (235)	Population-based mother–child cohort	ASQ, M-CHAT and IFDC	Confirmed by RT-PCR	Born to ZIKV-confirmed mothers. Only included infants who presented positive ZIKV serologies in cord and/or neonatal/infant blood	24 months (1)	Yes, children born to mothers with negative IgG at delivery

AIMS: Alberta Infant Motor Scale; ASQ: Ages and Stages Questionnaire; BSID-III: Bayley Scales of Infant and Toddler Development, Third Edition; DDST: Denver Developmental Screening Test, II Edition; FTII: Fagan Test of Infant Intelligence; GM: General Movement; IFDC: French MacArthur Inventory Scales; Ig: Immunoglobulin; M-CHAT: Modified Checklist for Autism on Toddlers; MSEL: Mullen Scales of Early Learning; PHENC: Public Health Emergency of National Concern; PRNT: plaque-reduction neutralization assay; RT-PCR: Reverse transcription polymerase chain reaction; SWYC: Survey of Wellbeing of Young Children; US: ultrasound; WIDEA: Warner Initial Developmental Evaluation of Adaptive and Functional Skills. * ZIKV definition based on CDC criteria; ** The study was conducted in Spain, but women were migrants who had recently travelled to an area of risk for ZIKV (Colombia, The Dominican Republic, etc.). *** Validated screening tools (https://www.aap.org/en-us/advocacy-and-policy/aap-health-initiatives/Screening/Pages/Screening-Tools.aspxexternalicon (accessed on 24 March 2021)).

**Table 2 ijerph-19-07319-t002:** Main results from the articles evaluating infant or child neurodevelopment with a tool other than BSID-III.

First Author	Characteristics: Children Evaluated (*n*), Inclusion Criteria, Tool Used and Main Study Objective	Socioeconomic Characteristics	Main Results	Neurodevelopment Affected in ZIKV Exposed Children
**Soares-Marangoni D.A.**	*n*= 2. Normocephalic children with negative serologies for other congenital infections, and negative ZIKV-PCR on urine, cerebrospinal fluid and umbilical cord samples; born to women with positive ZIKV-PCR. Prechtl’s General Movement assessment and the Alberta Infant Motor Scale. To describe the GMs in the fidgety * period and the motor performance of two infants who were exposed to ZIKV during distinct trimesters of gestation.	Not reported.	GMs in the fidgety period are early markers of motor performance at 12 months of age. In Case 1, fidgety movements were absent at 16 weeks after term and motor development was severely impaired at 12 months of age. In Case 2, fidgety movements were normal at 13 weeks and the motor outcome was typical at 12 months	Yes
**Rice M.E.**	*n* = 1386. Validated screening tools recommended by the American Academy of Pediatrics. Normocephalic children with no Zika-associated birth defects, born to women with laboratory evidence of confirmed or possible ZIKV infection. To report ZIKV-associated birth defects and/or neurodevelopmental abnormality possibly associated with congenital ZIKV, among one-year-old children born to mothers with confirmed or possible infection.	Not reported	Among 1450 children, 76% had developmental screening or evaluation, 60% had postnatal neuroimaging, 48% had automated auditory brainstem response-based hearing screen or evaluation, and 36% had an ophthalmologic evaluation. Among evaluated children, 6% had at least one ZIKV-associated birth defect, 9% had at least one neurodevelopmental abnormality possibly associated with congenital ZIKV infection, and 1% had both	Yes
**Oliveira Vianna R.A.**	*n* = 82. Children whose mothers had a rash and tested positive to ZIKV by PCR (Group 1); children whose mothers tested negative by PCR (Group 2); and children whose mothers did not undergo any testing for ZIKV but tested negative for other congenital infections (Group 3). DDST. To better understand the clinical spectrum and course of CZS during the first 18 months of life of children whose mothers had rash during pregnancy.	Most women were less than 30 years old, had at least 9 years of schooling; 37% of families earned one or less Brazilian monthly minimum wage, and 54% were residents of informal human settlements **	From the 108 children in the study, 26 developed CZS; thus, only 82 were healthy asymptomatic children. At 12 months, 7 of 82 children with no CZS (8.5%) had isolated abnormalities by physical examination that did not fulfil the criteria for CZS.	Yes
**Valdes V.**	*n* = 65. Children born to mothers with at least one prenatal or postnatal positive ZIKV-PCR. MSEL. To determine whether infants of mothers with at least one positive ZIKV test during pregnancy show differences in cognitive scores at ages 3 to 6 months and ages 9 to 12 months.	Mothers and fathers in the study had high levels of education (93.8%, and 75% with high school level education or higher, respectively), while 58.3% of mothers and 13.9% of fathers were unemployed or worked from home. Most of the children (88.9%) spoke Spanish, while the others were bilingual (Spanish and English). Regarding home status, 44.4% of families owned a house, 22.2% rented a house, 22.2% lived in public housing, and 11.1% occupied a house for free.	Prenatal maternal ZIKV infection is associated with lower receptive language scores during the first year of life; however, exposure to ZIKV does not appear to be associated with other domains of cognitive development. Maternal education, paternal education, maternal employment, paternal employment, and home status were tested to assess a possible association with ZIKV status. Maternal employment was the only variable significantly associated with ZIKV status (χ^2^ = 6.72; Cramér V = 0.32; *p* = 0.04).	Yes (only the language function)
**Lee E.H.**	*n*= 148 Children without birth defects, nor laboratory evidence of congenital ZIKV infection, born to women with laboratory evidence of ZIKV infection. No test specified. To characterize the epidemiology and clinical significance of congenital ZIKV exposure by prospectively following a cohort of infants with possible congenital exposure through their first year of life.	Not reported.	Most children, 95.3% (385), appeared well, whereas 19 (4.7%) had a possible ZIKV-associated birth defect. From 370 infants with neither birth defects nor laboratory evidence of congenital infection, or with no ZIKV testing, information at 12 months of age was available for 148 cases. Overall, 4 of 148 infants were reported to have a developmental delay; 2 infants demonstrated gross motor and speech delays, and 2 had isolated speech delay. Of the 22 infants younger than 12 months, only 13 had follow-up, and all of them had normal neurodevelopment.	Yes
**Mulkey S.B.**	*n* = 70 Normocephalic live-born with normal fetal brain findings on MRI, and average examination results without clinical evidence of CZS, born to women with laboratory evidence of ZIKV infection. WIDEA and AIMS. To assess the neurodevelopment of children exposed to ZIKV in utero born without CZS.	Not reported.	Infants with in utero ZIKV exposure without CZS appeared at risk for abnormal neurodevelopmental outcomes in the first 18 months of life. The WIDEA total score (coefficients: age = –0.227 vs. age2 = 0.006; *p* < 0.003) and self-care domain score (coefficients: age = –0.238 vs. age2 = 0.01; *p* < 0.008) showed curvilinear associations with age. Other domain scores showed linear declines with increasing age based on coefficients for communication (–0.036; *p* = 0.001), social cognition (–0.10; *p* < 0.001), and mobility (–0.14; *p* < 0.001). The AIMS scores were similar to the normative sample over time (95% CI, –0.107 to 0.037; *p* = 0.34). Overall, 19 of 57 infants (33%) who underwent postnatal cranial ultrasonography had a nonspecific, mild finding. No difference was found in the decline of WIDEA z scores between infants with and those without cranial ultrasonography findings except for a complex interactive relationship involving the social cognition domain (*p* < 0.049). The AIMS z scores were lower in infants with nonspecific cranial ultrasonography findings (–0.49; *p* = 0.07).	Yes
**Da Silva P.F.S.**	*n* = 140 Normocephalic children born to women with confirmed ZIKV-PCR. SWYC. To investigate patterns of neurodevelopment and behavior in groups of children with different severities of ZIKV-related microcephaly and children with prenatal ZIKV exposure in the absence of microcephaly.	Not reported.	ZIKV-exposed children without microcephaly and neurotypical controls had similar frequencies of risk of development delay. In comparison, 13.8% of ZIKV-exposed normocephalic children and 21.7% of control group children were identified by SWYC assessment as being ‘at risk’.	No
**Familiar I.**	*n* = 59 Normocephalic and asymptomatic children, born to women with confirmed ZIKV-PCR. MSEL and FTII. To assess neurodevelopmental outcomes in normocephalic infants born to women with ZIKV infection during pregnancy in Mexico.	Maternal educational level was high in the ZIKV-exposed group (95%), and in the unexposed group (89%). Overall, 78% and 80% of women were unemployed or worked from home in the ZIKV-exposed and -unexposed groups, respectively. No significant differences in demographic or anthropometric characteristics were observed.	All MSEL sub-scale scores, except expressive language, were significantly lower among ZIKV-exposed children compared to controls, including the overall standard composite (80 ± 10 vs. 87 ± 7.4, respectively; *p* < 0.001). In comparison with their peers, infants born to women with confirmed ZIKV infection during pregnancy showed poorer neurodevelopmental performance in language and motor domains and worse visual recognition memory at six months of age.	Yes
**Cabral Maia A.M.P.**	*n* = 17 Normocephalic children born to women with laboratory confirmed ZIKV infection. Child Health Booklet developed by the Brazilian Ministry of Health To evaluate the developmental and anthropometric milestones of asymptomatic children whose mothers had ZIKV infection.	Only one-third of mothers had completed high school (7/17, 41.2%); 7/17 (41.2%) were married, and 8/17 (47.1%) were housewives. The average income was one minimum wage (954.00 BRL). Among the women who were housewives, 3/8 (37.5%) had quit their jobs to take care of their children.	Most children, 15/17 (88.2%), presented with at least one delayed developmental milestone with respect to the standards for the age group. Among these children, 5/15 (33.3%) reached three developmental milestones, 5/15 (33.3%) reached two, and 5/15 (33.3%) reached only one.	Yes
**Pimentel R.**	*n* = 42 Children born without obvious ZIKV-associated birth defects, to symptomatic women with confirmed ZIKV-PCR.DDST. To assess the clinical and epidemiological characteristics of infants with ZIKV-associated microcephaly, and the neurodevelopmental abnormalities during the first 18 months of life for a group of infants with possible congenital ZIKV exposure.	Authors compared sociodemographic characteristics and clinical presentation of mothers with or without an infant with abnormal developmental screening, and found no significant differences between the two groups except for higher-frequency abdominal pain during pregnancy in women whose infants had an abnormal developmental screen (85% vs. 38%, *p* = 0.007). Although the sample size is small, maternal alcohol use and smoking history were not associated with infant’s developmental delay.	Of 42 infants with possible congenital ZIKV exposure followed longitudinally, 52% exhibited possible developmental delay in at least one visit throughout the 18-month observation period. Interestingly, most of the observed neurodevelopmental abnormalities resolved over time and only four infants were noted to have abnormalities that persisted for 15–18 months. If the two infants who developed postnatal microcephaly were excluded, 5% (2/42) of infants had neurodevelopmental abnormalities possibly associated with congenital ZIKV infection.	Yes
**Grant R.**	*n*= 235 Normocephalic children born with normal transfontanelle cerebral ultrasound findings, or normal ultrasound findings on the last ultrasound performed during the third trimester of the mother’s pregnancy, born to women with confirmed ZIKV-PCR. ASQ, M-CHAT, and IFDC. To determine the impact of ZIKV exposure on neurodevelopment at 24 months of age among toddlers who were born normocephalic to women who were pregnant during the 2016 ZIKV outbreak in French territories in the Americas.	Comparisons between ZIKV-exposed and unexposed toddlers indicated a lower maternal age (*p*= 0.01), higher maternal education (*p* = 0.04), and higher paternal education (*p* = 0.04) in the unexposed; a higher proportion of toddlers from Guadeloupe in the exposed group and a higher proportion of toddlers from Martinique in the unexposed group (*p*≤ 0.001); higher parity in the ZIKV exposed (*p* = 0.04); and greater use of mosquito repellents in the exposed group (*p* = 0.05).	In one of the largest population-based, mother–child cohorts of in utero ZIKV-exposed normocephalic at birth to date. Authors found that 15.3% of toddlers exposed to ZIKV have abnormal neurodevelopment findings at 24 months of age. However, differences were not statistically significant when compared to not-exposed toddlers.	No

AIMS: Alberta Infant Motor Scale; ASQ: Ages and Stages Questionnaire; BSID-III: Bayley Scales of Infant and Toddler Development, Third Edition; CZS: Congenital Zika Syndrome; DDST: Denver Developmental Screening Test, II Edition; FTII: Fagan Test of Infant Intelligence; GM: General Movement; IFDC: French MacArthur Inventory Scales; M-CHAT: Modified Checklist for Autism on Toddlers; MSEL: Mullen Scales of Early Learning; SWYC: Survey of Wellbeing of Young Children; WIDEA: Warner Initial Developmental Evaluation of Adaptive and Functional Skills. ZIKV: Zika virus; * Infants with normal fidgety movements at 3 to 5 months are very likely to show neurologically normal development; ** Designated by the Brazilian Institute of Geography and Statistics as aglomerados subnormais (AGSN).

**Table 3 ijerph-19-07319-t003:** Disaggregated data, prevalence, and weight of each study in the meta-analysis for any type of delay in the cognitive, language, and motor domains using the BSID-III scales.

First Author	Number of Children Evaluated	Cognitive Delay			Language Delay			Motor Delay		
		Affected Children	Prevalence (95% CI)	% Weight	Affected Children	Prevalence (95% CI)	% Weight	Affected Children	Prevalence (95% CI)	% Weight
**Nielsen-Saines K.**	146	14	9.6 (5.8, 15.5)	30.49	51	34.9 (27.7, 43.0)	16.73	24	16.4 (11.3, 23.3)	14.78
**Lopes Moreira M.E.**	94	11	11.7 (6.7, 19.8)	19.67	25	26.6 (18.7, 36.3)	15.43	18	19.1 (12.5, 28.3)	14.12
**Peçanha P.M.**	84	4	4.8 (1.9, 11.6)	17.59	31	36.9 (27.4, 47.6)	15.05	20	23.8 (16.0, 33.9)	13.92
**Einspieler C.**	56	1	1.8 (0.3, 9.4)	11.76	6	10.7 (5.0, 21.5)	13.46	1	1.8 (0.3, 9.4)	13.03
**Faiçal A.V.**	29	4	13.8 (5.5, 30.6)	6.14	9	31.0 (17.3, 49.2)	10.41	1	3.4 (0.6, 17.2)	11.08
**Abtibol-Bernardino M.R.**	26	5	19.2 (8.5, 37.9)	5.52	10	38.5 (22.4, 57.5)	9.89	9	34.6 (19.4, 53.8)	10.70
**Marbán-Castro E. ****	21	1	4.8 (0.8, 22.7)	4.47	7	33.3 (17.2, 54.6)	8.86	0	0.0 (0.0, 15.5)	9.92
**Rodrigues Gerzson L.**	17	1	5.9 (1.0, 27.0)	3.64	9	52.9 (31.0, 73.8)	7.86	4	23.5 (9.6, 47.3)	9.12
**Sobhani N.C.**	3	0	0.0 (0.0, 56.1)	0.73	0	0.0 (0.0, 56.1)	2.31	0	0.0 (0.0, 56.1)	3.33
**Pooled prevalence ***		6.5 (4.1, 9.3) Q Heterogeneity chi-squared = 11.60 (d.f. = 8) *p* = 0.1701; I^2^ = 31.0%			29.7 (0.217, 0.382) Q Heterogeneity chi-squared = 22.77 (d.f. = 8) *p* = 0.0037; I^2^: = 64.9%			11.5 (4.8, 20.1) Q Heterogeneity chi-squared = 37.02 (d.f. = 8) *p* <0.0001; I^2^ = 78.4%		

I^2^: variation in prevalence attributable to heterogeneity. * Cognitive domain: Differences between publications are low (I^2^ = 31.0%); thus, fixed-effects pooled prevalence was considered. Language domain: Differences between publications are large (I^2^ = 64.9%); thus, random-effects pooled prevalence was considered. Motor domain: Differences between publications are large (I^2^ = 78.4%); thus, random-effects pooled prevalence was considered. ** All of these studies were conducted in Brazil, except the one from Marbán-Castro et al., which was conducted in Spain.

**Table 4 ijerph-19-07319-t004:** Disaggregated data, prevalence, and weight of each study in the meta-analysis for moderate and severe delay in the cognitive, language, and motor domains using the BSID-III scales.

First Author	Number of Children Evaluated	Moderate and Severe Cognitive Delay			Moderate and Severe Language Delay			Moderate And Severe Motor Delay		
		Affected Children	Prevalence (95% CI)	% Weight	Affected Children	Prevalence (95% CI)	% Weight	Affected Children	Prevalence (95% CI)	% Weight
**Nielsen-Saines K.**	146	8	5.5 (2.8, 10.4)	36.99	17	11.6 (7.4, 17.9)	36.99	7	4.8 (2.3, 9.6)	36.99
**Lopes Moreira M.E.**	94	6	6.4 (3.0, 13.2)	23.86	10	10.6 (5.9, 18.5)	23.86	7	7.4 (3.7, 14.6)	23.86
**Einspieler C.**	56	0	0.0 (0.0, 6.4)	14.27	3	5.4 (1.8, 14.6)	14.27	1	1.8 (0.3, 9.4)	14.27
**Faiçal A.V.**	29	4	13.8 (5.5, 30.6)	7.45	2	6.9 (1.9, 22.0)	7.45	0	0.0 (0.0, 11.7)	7.45
**Abtibol-Bernardino M.R.**	26	0	0.0 (0.0, 12.9)	6.69	6	23.1 (11.0, 42.1)	6.69	3	11.5 (4.0, 29.0)	6.69
**Marbán-Castro E. ****	21	0	0.0 (0.0, 15.5)	5.43	1	4.8 (0.8, 22.7)	5.43	0	0.0 (0.0, 15.5)	5.43
**Rodrigues Gerzson L.**	17	0	0.0 (0.0, 18.4)	4.42	2	11.8 (3.3, 34.3)	4.42	0	0.0 (0.0, 18.4)	4.42
**Sobhani N.C.**	3	0	0.0 (0.0, 56.1)	0.88	0	0.0 (0.0, 56.1)	0.88	0	0.0 (0.0, 56.1)	0.88
**Pooled prevalence ***		1.9 (0.4, 4.1) Q Heterogeneity chi-squared = 13.42 (d.f. = 7) *p* = 0.0626; I^2^ = 47.8%			8.4 (5.4, 11.8) Q Heterogeneity chi-squared = 6.08 (d.f. = 7) *p* = 0.5306; I^2^= 0.0%			2.2 (0.6, 4.5) Q Heterogeneity chi-squared = 8.49 (d.f. = 7) *p* = 0.2918; I^2^ = 17.5%		

I^2^: variation in prevalence attributable to heterogeneity. * Cognitive domain: Differences between publications are low (I^2^ = 47.8%), so fixed-effects pooled prevalence was considered. Language domain: Differences between publications are low (I^2^ = 0.0%); thus, fixed-effects pooled prevalence was considered. Motor domain: Differences between publications are low (I^2^ = 17.5%); thus, fixed-effects pooled prevalence was considered. ** All of these studies were conducted in Brazil, except the one from Marbán-Castro et al., which was conducted in Spain.

## Data Availability

No new data were created or analyzed in this study. Data sharing is not applicable to this article.

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
