# Peer review of "Neurodevelopment in Normocephalic Children Exposed to Zika Virus in Utero with No Observable Defects at Birth: A Systematic Review with Meta-Analysis"

_ijerph, 2022, doi:10.3390/ijerph19127319_

Round 1

Reviewer 1 Report

Re: ijerph-1705777

Marbán-Castro et al have written a review and meta-analysis of neurodevelopmental abnormalities with in utero Zika exposure, outside the setting of microcephaly.  This is of interest.  However, it might be good to go ahead and make some mention of the data on the incidence of microcephaly, just for context, and highlight the clinical context- that microcephaly is readily apparent at birth, so it is critical to know what the risks are for those without such a readily apparent issue at birth.  With the many challenges in confirmation of Zika mentioned in the 2nd paragraph, I was wondering as to the authors’ confidence in the overall results- but the authors handled this concern quite well in the Discussion.  I do have a few other comments. 

In the Abstract, and again later in the paper, it seems the authors should clarify that ‘cognitive delay’ refers specifically to non-language cognitive delays.  After all, language is an aspect of cognition, so it looks rather surprising to otherwise see that cognitive delay is less common that its potential subtype: language delay.

Introduction- change ‘does not present’ to ‘do not present’.    What is the salience of the chest ultrasound mentioned?  Seems out of context here.

Methods- After the detailed search decision tree, the authors add mention of a supplemental search with 4 specific key words.  Could the authors clarify what was done here?  This was particularly odd after the high specificity of the decision tree.  Later ‘Studies including data from children with microcephaly…were excluded’- do the authors mean to clarify that these were excluded only in cases where the microcephaly (or other abnormalities observed at birth) could not be separated from the children for whom this was not the case?  Later, it is clear that there are at least a couple of studies included where there were microcephalic cases, but the ones without microcephaly could clearly be tracked separately.  Might have a paragraph break or another mechanism to distinguish ‘For the meta-analysis, addition exclusion criteria were applied…’- as it is currently buried in the middle of the paragraph, and the reader might suddenly wonder why only the BSID-III is being used in the review.  It makes sense that the meta-analysis is limited to this assessment.  What is meant by ‘and those response gradient’ in the risk of bias assessment? 

Results- Might clarify for the reader how there were 22 articles from 20 studies- did 2 studies separately publish one early and one late follow-up study each?  Table 1- one of the studies has nothing listed under Maternal and infant-status related inclusion criteria (infants)- please clarify (Soarres-Marangoni et al).  Is there any further information on what are the ‘Validated screening tools recommended by the American Academy of Pediatrics’ (Rice et al- in Table 2 as well)?  And for Oliveira Vianna et al, ‘No’ is stated under Control group, but one of the groups was ‘Born to ZIKV negative mothers’- is that not a control group? Clarify ‘abnormal brain imaging was detected in children from two studies’- does that just mean that two studies also had a microcephaly group in addition to non-microcephalic?  There seems to be something missing with ‘Selection of study participants and definition of unexposed and exposed pregnant women and infants’.  There appears to be 2 Table S1s, and the 2nd one doesn’t seem to be called.  Also, there is a call for a Table S2, which does not exist, and it comes after the call for Table S3.  Please check on the supplementary tables.  Table 2- what is ‘Child’s booklet’ mentioned under tool used?

Discussion- what is meant by ‘particularly in the fidgety period’?   For ‘with transmission of several arboviruses and infection may be subclinical’- do the authors mean to add ‘where’ before ‘infection’?  Also for ‘which might have increased the number of children in this group’- do the authors mean ‘this’ instead of ‘which’?

Author Response

Reviewer 1 comment: Marbán-Castro et al have written a review and meta-analysis of neurodevelopmental abnormalities with in utero Zika exposure, outside the setting of microcephaly.  This is of interest.  However, it might be good to go ahead and make some mention of the data on the incidence of microcephaly, just for context, and highlight the clinical context- that microcephaly is readily apparent at birth, so it is critical to know what the risks are for those without such a readily apparent issue at birth.  With the many challenges in confirmation of Zika mentioned in the 2nd paragraph, I was wondering as to the authors’ confidence in the overall results- but the authors handled this concern quite well in the Discussion.  I do have a few other comments. 

Authors’ response: Thank you very much for your appreciation of our work. Following your advice, we have included some of the scarce data on the incidence of Zika-associated microcephaly for context.

Introduction: Page 2, line 49: “Fetuses of infected women have a 5 to 14% risk of developing CZS, and a 4 to 6% risk of microcephaly; and those who develop CZS, 80% of them present microcephaly [5].”

Indeed, the risks for those without readily apparent microcephaly at birth is the focus of the review.

Reviewer 1 comment:In the Abstract, and again later in the paper, it seems the authors should clarify that ‘cognitive delay’ refers specifically to non-language cognitive delays.  After all, language is an aspect of cognition, so it looks rather surprising to otherwise see that cognitive delay is less common that its potential subtype: language delay.

Authors’ response: Thank you very much for such detail. We followed the terms used by the neurodevelopmental tool (BSID-III). Surely, we agree with language being a cognitive delay, however it has been analysed and reported separately, because of its importance. We have clarified that definition in the methods section, and added that clarification in the Abstract as “non-language cognitive delay”.

Materials and methods: Page 3, lines 116-119: “In this article, definition of “cognitive domain/delay” is based on the neurodevelopmental tool used. While “language” is an aspect of cognition, when we mention “cognitive” we refer to “non-language domain/delay”.

Reviewer 1 comment: Introduction- change ‘does not present’ to ‘do not present’.    What is the salience of the chest ultrasound mentioned?  Seems out of context here.

Authors’ response: Thank you very much for realizing these errors. Those have been addressed. The word “chest” has been deleted.

Reviewer 1 comment: Methods- After the detailed search decision tree, the authors add mention of a supplemental search with 4 specific key words.  Could the authors clarify what was done here?  This was particularly odd after the high specificity of the decision tree.  

Authors’ response: Thank you for your response. Indeed, the search string included in the manuscript was correct (page 3, lines 107-110) “The search string included the following text terms: ((Zika OR ZIKV) AND (Child OR children OR infant) AND (neurodevelopment OR neurodevelopmental OR development OR neurodevelopmental OR Bayley OR BSID OR delay OR language OR motor OR cognitive)), see TableS1 in the supplementary material.” That other sentence mentioned about these four words might have been a research note, that was erroneously integrated in the manuscript. I confirm that now the terms in the search included are correct.

Reviewer 1 comment: Later ‘Studies including data from children with microcephaly…were excluded’- do the authors mean to clarify that these were excluded only in cases where the microcephaly (or other abnormalities observed at birth) could not be separated from the children for whom this was not the case?  Later, it is clear that there are at least a couple of studies included where there were microcephalic cases, but the ones without microcephaly could clearly be tracked separately.  Might have a paragraph break or another mechanism to distinguish ‘For the meta-analysis, addition exclusion criteria were applied…’- as it is currently buried in the middle of the paragraph, and the reader might suddenly wonder why only the BSID-III is being used in the review.  It makes sense that the meta-analysis is limited to this assessment.  What is meant by ‘and those response gradient’ in the risk of bias assessment? 

Authors’ response: Yes, we edited the text to be clearer on this regard. Page 3, lines 122-125: “Studies including only data from children with microcephaly, or other ZIKV-associated adverse outcomes or conditions that were observable at birth, were excluded. Studies that also included data on children with no-observable defects are birth were included.”

Thank you for the suggestion to break the paragraph about the meta-analysis. It is clearer now.

Reviewer 1 comment: Results- Might clarify for the reader how there were 22 articles from 20 studies- did 2 studies separately publish one early and one late follow-up study each?  

Authors’ response: There were 22 articles published with analysis and sub-analysis of the cohorts, however two articles were performed within 2 other studies. One sentence has been added for clarity in page 4 lines 184-186 “Two articles included in this review were performed with children from the same study from different time points.”.

Reviewer 1 comment: Table 1- one of the studies has nothing listed under Maternal and infant-status related inclusion criteria (infants)- please clarify (Soarres-Marangoni et al).  

Authors’ response: Thank you for realizing that, and sorry for the mistake. The text has been entered accordingly in the Table as “Born to ZIKV con-firmed mothers. Infants not tested for ZIKV”.

Reviewer 1 comment: Is there any further information on what are the ‘Validated screening tools recommended by the American Academy of Pediatrics’ (Rice et al- in Table 2 as well)?  

Authors’ response: The article by Rice et al included the following information “Standard evaluation includes a comprehensive physical exam, including growth parameters; newborn hearing screen, preferably with automated auditory brainstem response (ABR); developmental monitoring and screening using validated screening tools recommended by the American Academy of Pediatrics (https://www.aap.org/en-us/advocacy-and-policy/aap-health-initiatives/Screening/Pages/Screening-Tools.aspxexternal icon); and vision screening as recommended by the American Academy of Pediatrics Policy Statement “Visual System Assessment in Infants, Children, and Young Adults by Pediatricians” (http://pediatrics.aappublications.org/content/137/1/e20153596external icon).”.

However we did not include this information to minimize the content of the manuscript, which has already lots of information.

Reviewer 1 comment: And for Oliveira Vianna et al, ‘No’ is stated under Control group, but one of the groups was ‘Born to ZIKV negative mothers’- is that not a control group?

Authors’ response: Yes, thank you very much for highlighting that. It was a mistake while transcribing the data to the tables.

Reviewer 1 comment: Clarify ‘abnormal brain imaging was detected in children from two studies’- does that just mean that two studies also had a microcephaly group in addition to non-microcephalic?  

Authors’ response: We have included “structural and nonstructural abnormalities” from the article from Lopes-Moreira et al; and calcifications and cysts from Marbán-Castro et al; as that was what authors stated. We included “abnormal brain imaging, including structural and nonstructural abnormalities, calcifications and cysts” to clarify it in line 213.

Reviewer 1 comment: There seems to be something missing with ‘Selection of study participants and definition of unexposed and exposed pregnant women and infants’.  There appears to be 2 Table S1s, and the 2nd one doesn’t seem to be called.  

Authors’ response: Sorry for the misunderstanding. There are 4 Tables in the Supplementary Materials; as stated at the end of the manuscript:

Table S1. Systematic review search strategy;

Table S2: Full data on the prevalence of neurodevelopmental delays in children with prenatal ZIKV exposure and study authors;

Table S3: Neurodevelopmental tools chosen to measure the long-term impact of prenatal ZIKV exposure in children;

Table S4: Assessment of the quality of the articles included in the meta-analysis (studies evaluating neurodevelopment with the BSID-III scale).

Reviewer 1 comment: Also, there is a call for a Table S2, which does not exist, and it comes after the call for Table S3.  Please check on the supplementary tables.  

Authors’ response: Thank you for highlighting this. There are 4 tables in the supplementary material Tables S1, S2, S3, and S4. The title of Table S2 was erroneously mentioned as “S1” in the supplementary material. Now it has been reviewed and edited accordingly.

Reviewer 1 comment: Table 2- what is ‘Child’s booklet’ mentioned under tool used?

Authors’ response: Thank you for this comment. We have added “Child Health Booklet developed by the Brazilian Ministry of Health” to be more self-explicative.

Reviewer 1 comment: Discussion- what is meant by ‘particularly in the fidgety period’?   

Authors’ response: The fidgety period is defined as the movement period during the post-term age of 3–5 months, were spontaneous movement patterns typically occurs.

Reviewer 1 comment: For ‘with transmission of several arboviruses and infection may be subclinical’- do the authors mean to add ‘where’ before ‘infection’?  

Authors’ response: Thank you for realizing that. We have added “where” accordingly.

Reviewer 1 comment: Also for ‘which might have increased the number of children in this group’- do the authors mean ‘this’ instead of ‘which’?

Authors’ response: Thank you, we have changed the word to “which”.

Thank you very much for your appreciation of our work, and all the detailed comments. We have addressed all of them in the new version of the manuscript.

Reviewer 2 Report

Well written article.

some minor corrections.

line 47/48 does to be replaced by do

line 90 add is after review

Author Response

Thank you very much for your corrections.

Reviewer 3 Report

The Manuscript follows the systematic review protocols according to PRISMA, including previous records in PROSPERO. This is important because there are reviews prior to the years selected in this manuscript. It would be important to include in the introduction the justification for this systematic realization in these terms. In methodology, there is a well-defined enumeration of it. The question is whether the reviews considered these inclusion criteria or whether the present manuscript sheds new light on the analysis of the subject, especially highlighting that one of the foundations of the review is to examine the effect of the incidence of this virus in children. long-term.

In any case, the methodology is well exposed and the results express what is relevant regarding the findings. It is suggested to highlight that it is one of the first studies in this regard, taking care that there are other databases that could contain publications in this regard.

Although most of the antecedents of behavioral and sociocognitive deterioration are exposed and are in accordance with problems detected early, I suggest careful wording regarding the conclusion, especially when referring to language. It is not clear, at least in the particular review I have done, that the language delay is only due to the virus. The reason for this is that at 24 months (2 years of age), with a dispersion between 15 and 30 months, the development of this function may be delayed due to other factors, including nutrition, and socioeconomic status, among others. Neuroscientific studies have shown that this function may be delayed. The suggestion is that this conclusion is carefully referred to, leaving the possibility of other causes that were not part of this study. On the other hand, the cognitive aspects present a wide dispersion in children under 6 years of age and with greater reason in children under 3 years, as to be conclusive in this regard.

However, in general, the manuscript is of ample quality.

Author Response

Reviewer 3 comment: The Manuscript follows the systematic review protocols according to PRISMA, including previous records in PROSPERO. This is important because there are reviews prior to the years selected in this manuscript. It would be important to include in the introduction the justification for this systematic realization in these terms.

Authors’ response: Thank you very much for your recognition of our work. Following your advice we have included a sentence in the introduction.

Page 2, lines 92-93: This systematic review was conducted based on the lack of evidence on health effects of children born from ZIKV infected mothers, with no observable birth defects.

Reviewer 3 comment: In methodology, there is a well-defined enumeration of it. The question is whether the reviews considered these inclusion criteria or whether the present manuscript sheds new light on the analysis of the subject, especially highlighting that one of the foundations of the review is to examine the effect of the incidence of this virus in children. long-term.

Authors’ response: Thank you for this comment. This systematic review sheds new light on the analysis of the subject, as it includes data from different studies conducted with various populations, but to our concern, is the first systematic review performed on this topic.

Reviewer 3 comment: In any case, the methodology is well exposed and the results express what is relevant regarding the findings. It is suggested to highlight that it is one of the first studies in this regard, taking care that there are other databases that could contain publications in this regard.

Authors’ response: Thank you very much for highlighting this. A sentence was written in the conclusions section addressing this. Hope that it was what reviewer expected.

Page 20, lines 465-467: “This is the first study assessing the pooled prevalence of neurodevelopmental delays estimated in studies of normocephalic children prenatally exposed to ZIKV.”

Reviewer 3 comment: Although most of the antecedents of behavioral and sociocognitive deterioration are exposed and are in accordance with problems detected early, I suggest careful wording regarding the conclusion, especially when referring to language. It is not clear, at least in the particular review I have done, that the language delay is only due to the virus. The reason for this is that at 24 months (2 years of age), with a dispersion between 15 and 30 months, the development of this function may be delayed due to other factors, including nutrition, and socioeconomic status, among others. Neuroscientific studies have shown that this function may be delayed. The suggestion is that this conclusion is carefully referred to, leaving the possibility of other causes that were not part of this study. On the other hand, the cognitive aspects present a wide dispersion in children under 6 years of age and with greater reason in children under 3 years, as to be conclusive in this regard.

Authors’ response: Thank you very much for this detailed response. Indeed, we understand that delays are multifactorial and might be caused by other factors. We have added another sentence in the conclusions to make them clearer:

Page 20, lines 475-477: “Delayed child neurodevelopment might be due to different factors, such as nutrition or socioeconomic factors. Those factors might confound the association of ZIKV prenatal exposure with neurodevelopmental delays and adjustment for these factors, as through matched controls, should be included in future studies.”

Reviewer 3 comment: However, in general, the manuscript is of ample quality.

Authors’ response: Thank you very much for your appreciation of our work.

Round 2

Reviewer 1 Report

Re: ijerph-1705777v2

Marbán-Castro et al have written a review and meta-analysis of neurodevelopmental abnormalities with in utero Zika exposure, outside the setting of microcephaly.  This is of interest.  The authors have addressed most of my issues, but a few remain.

Would highlight the clinical context somewhere in the paper- that microcephaly is readily apparent at birth, so it is critical to know what the risks are for those without such a readily apparent issue at birth.

Introduction- new text change ‘present microcephaly’ to ‘present with microcephaly’

Methods- What is meant by ‘and those response gradient’ in the risk of bias assessment? 

Results- Table 1- for Oliveira Vianna et al, under Control group, change ‘26children’ to ’26 children’.  There seems to be something missing with ‘Selection of study participants and definition of unexposed and exposed pregnant women and infants’- this is not a full sentence.  The call for Table S2 comes after the call for Table S3-just the order is odd.  Somewhere, give some indication of what makes up the ‘Validated screening tools recommended by the American Academy of Pediatrics’- should indicate this somewhere in the paper.

Discussion- what is meant by ‘particularly in the fidgety period’?- this should be described somewhere in the paper.  

Author Response

Reviewer 1 comment: Marbán-Castro et al have written a review and meta-analysis of neurodevelopmental abnormalities with in utero Zika exposure, outside the setting of microcephaly.  This is of interest.  The authors have addressed most of my issues, but a few remain.

Would highlight the clinical context somewhere in the paper- that microcephaly is readily apparent at birth, so it is critical to know what the risks are for those without such a readily apparent issue at birth.

Authors’ response: Thank you for that suggestion. We have integrated that sentence in page 2, lines 51-53: “Microcephaly is usually readily apparent at birth, but Some adverse consequences of in utero ZIKV exposure may only be observable in early childhood or when children start school, what is still critical to know [6].”

Reviewer 1 comment: Introduction- new text change ‘present microcephaly’ to ‘present with microcephaly’

 Authors’ response: Thank you very much. That word has been added accordingly.

Reviewer 1 comment: Methods- What is meant by ‘and those response gradient’ in the risk of bias assessment? 

Authors’ response: Thank you for realising this. The words should be “in the response gradient” because that correspond to the risk of bias assessment (by colour depending on risks). But it is true that might be confusing. We have removed those words and left in page 4, lineb156“others (inconsistency, indirectness, imprecision, publication bias, large effect, and possible confounding)”.

Reviewer 1 comment: Results- Table 1- for Oliveira Vianna et al, under Control group, change ‘26children’ to ’26 children’.  

Authors’ response: Changed! Thanks for such a detailed look!

Reviewer 1 comment: There seems to be something missing with ‘Selection of study participants and definition of unexposed and exposed pregnant women and infants’- this is not a full sentence.  

Authors’ response: Thank you for this comment. Indeed, we have deleted this sentence, full selection of participants and definitions are detailed in the following sub-headings, but that heading was not necessary.

Reviewer 1 comment: The call for Table S2 comes after the call for Table S3-just the order is odd.  

Authors’ response: Thank you for realizing, we have inverted the order of those tables accordingly.

Reviewer 1 comment: Somewhere, give some indication of what makes up the ‘Validated screening tools recommended by the American Academy of Pediatrics’- should indicate this somewhere in the paper.

Authors’ response: For reference to the readers we have added as a footnote in page 9, lines 201-202 “ ***Validated screening tools (https://www.aap.org/en-us/advocacy-and-policy/aap-health-initiatives/Screening/Pages/Screening-Tools.aspxexternal icon)”

Reviewer 1 comment: Discussion- what is meant by ‘particularly in the fidgety period’?- this should be described somewhere in the paper.  

Authors’ response: Following reviewers’ advice, the following sentence has been added as a footnote in Table 2, page 4, line 263-264, “ * Infants with normal fidgety movements at 3 to 5 months are very likely to show neurologically normal development”

Thank you very much for all your comments, that have helped us improve the quality of our manuscript.